# Development of a Promising ^18^F-Radiotracer for PET Imaging Legumain Activity In Vivo

**DOI:** 10.3390/ph15050543

**Published:** 2022-04-27

**Authors:** Chunmei Lu, Xiuting Wang, Qiqi Wang, Lixia Zhang, Jianguo Lin, Ling Qiu

**Affiliations:** 1School of Chemical and Material Engineering, Jiangnan University, Wuxi 214122, China; 6190606015@stu.jiangnan.edu.cn (C.L.); wang1xiuting@163.com (X.W.); garatiii@163.com (Q.W.); 2NHC Key Laboratory of Nuclear Medicine, Jiangsu Key Laboratory of Molecular Nuclear Medicine, Jiangsu Institute of Nuclear Medicine, Wuxi 214063, China; zhanglixia1017@126.com

**Keywords:** legumain, intramolecular condensation, ^18^F-labeling, positron emission tomography

## Abstract

Legumain has been found overexpressed in several cancers, which serves as an important biomarker for cancer diagnosis. In this research, a novel fluorine-18 labeled radioactive tracer [^18^F]**SF-AAN** targeting legumain was designed and synthesized for positron emission tomography (PET) imaging. Nonradioactive probe [^19^F]**SF-AAN** was obtained through chemical and solid phase peptide synthesis. After a simple one-step ^18^F labeling, the radiotracer [^18^F]**SF-AAN** was obtained with a high radiochemical conversion rate (>85%) and radiochemical purity (99%) as well as high molar activity (12.77 ± 0.50 MBq/nmol). The targeting specificity of [^18^F]**SF-AAN** for detecting legumain activity was investigated systematically in vitro and in vivo. In vitro cellular uptake assay showed that the uptake of [^18^F]**SF-AAN** in legumain-positive MDA-MB-468 cells was twice as much as that in legumain-negative PC-3 cells at 4 h. In vivo PET imaging revealed that the tumor uptake of [^18^F]**SF-AAN** in MDA-MB-468 tumor-bearing mice was about 2.7 times of that in PC-3 tumor-bearing mice at 10 min post injection. The experimental results indicated that [^18^F]**SF-AAN** could serve as a promising PET tracer for detecting the legumain expression sensitively and specifically, which would be beneficial for the diagnosis of legumain-related diseases.

## 1. Introduction

Cancer is one of the most important elements causing human death. According to the data reported by the World Health Organization (WHO), nearly 10 million people died of cancer worldwide in 2020 [1,2]. Given that the patients were diagnosed in the early stage of cancer, the cure rate would be significantly improved. Hence, early diagnosis could be made possible utilizing the simple and rapid detection of different biomarkers in organism, which is important for promoting the cure rate and survival rate of patients [3]. It is essential to obtain precise detection and location of cancer via choosing a specific biomarker for one target disease [4]. Enzymes play a critical role in life activities, reflecting the health status of living subjects which are related to carcinogenesis, so they become crucial indicators for early diagnosis of cancer [5].

Legumain, a cysteine protease, mainly exists in the lysosome under acidic environment [6,7,8,9]. The up-regulation of legumain is highly associated with cancer invasion, metastasis and angiogenesis [10]. According to reports, legumain is overexpressed at a quite high level in several cancers [11], including breast cancer [12,13,14,15], colorectal cancer [11] and gastric cancer [16]. Because legumain participates in tumor development, legumain has been regarded as an essential biomarker for the tumor diagnosis [7]. Since cancer is supposed to be potentially cured at its earliest stage [17,18], accurate determination of legumain activity and precise localization of tumor become very vital for early cancer diagnosis.

So far, a large number of probes have been reported for the detection of legumain [6,11,13,19,20,21,22,23,24]. In 2015, Liang and co-workers reported a nuclear magnetic resonance (NMR) probe for imaging legumain in HEK293T tumor-bearing zebrafish [20]. Then, they further developed a near-infrared fluorescence (NIRF) probe for detecting the legumain in HCT116 tumor-bearing mice in 2018 [11]. In addition, Gao et al. developed a gold nanoparticle to detect the legumain activity in orthotopic C6 glioma-bearing mice utilizing photoacoustic imaging [25]. Although the expression level of legumain in vivo could be visualized in real time via NMR, NIRF, or photoacoustic imaging, these imaging methods still have certain shortcomings compared with positron emission tomography (PET) imaging, such as limited depth of tissue penetration [26]. PET has been applied for detecting diseased tissues attributing to its higher depth of tissue penetration and sensitivity [27,28,29]. Fluorine-18 is a commonly used nuclide in clinical for its appropriate half-life (109.7 min) and plays an important part in drug exploitation and PET imaging [30]. However, fluorine-18 labeled PET imaging tracer targeting legumain was rarely explored. Recently, our group reported a PET imaging tracer **^18^F-2** for detecting legumain activity in HCT116 tumor-bearing mice [24]. The chemical structure of **^18^F-2** contained (1) a 6-amino-2-cyanobenzothiazole (CBT) structure and a disulfided cysteine (Cys) motif for rapid CBT-Cys condensation; (2) a legumain-recognition substrate (Ala-Ala-Asn); (3) a propargyl glycine for attaching the labeling group AMBF_3_ motif. However, this tracer needed to be co-injected with a nonradioactive probe for self-assembly to obtain ideal tumor uptake, which might induce a toxic and side effects on other normal organs. At the same time, the signal-to-noise ratio (SNR) was not high enough. Therefore, it is necessary to further optimize and design novel ideal PET tracers for accurately detecting the legumain activity in living subjects.

Inspired by intramolecular condensation and macrocyclization of probes [31,32], we intended to optimize the structure of **^18^F-2** to increase the tumor uptake and prolong the retention time of radioactivity at target site. On one hand, the introduction of glycine could increase the flexibility of molecular probe [33], in favor of intramolecular condensation. On the other hand, the introduction of rigid aromatic functional groups [32], such as 4-(aminomethyl)benzoic acid, could promote the assembly of intramolecular cyclic products through hydrophobic and π-π interactions. Hence, a new tracer [^18^F]**SF-AAN** was designed and synthesized based on the chemical framework **SF-11** [34] in the present work. According to a series of in vitro and in vivo biological evaluations, [^18^F]**SF-AAN** was used to detect the expression level of endogenous legumain to verify the targeting specificity of radiotracer. This would contribute to the early diagnosis and later treatment of cancer patients.

## 2. Results

### 2.1. Chemical Synthesis and Characterization

On the basis of reported tracer **^18^F-2**, we design and synthesize a novel tracer [^18^F]**SF-AAN** targeting legumain (Figure 1). Two 4-(aminomethyl)benzoic acids and a glycine was introduced to the chemical skeleton of **^18^****F-2** for further optimization as **SF-11**, which could perform intramolecular condensation. The legumain-recognition substrate and the labeling group were still Ala-Ala-Asn and AMBF_3_ motif.

The nonradioactive probe [^19^F]**SF-AAN** was synthesized according to the procedure illustrated in Appendix A. Firstly, the skeleton **SF-11** was synthesized according to the reported method [34]. Then, **SF-11** reacted with **A****c****-AAN-OH** to yield **SF-12** through a condensation. Subsequently, the protection group Trt was removed to yield **SF-13**. Finally, AMBF_3_ was attached to the alkynyl group of **SF-13** via a Cu(I) catalyzed click reaction to yield [^19^F]**SF-AAN** after high-performance liquid chromatography (HPLC) purification. All the compounds were characterized and confirmed by HPLC, electrospray ionization mass spectrometry (ESI-MS) and nuclear magnetic resonance (^1^H/^13^C NMR) (Appendix A), indicating that [^19^F]**SF-AAN** was successfully synthesized and the purity of [^19^F]**SF-AAN** was more than 97%.

### 2.2. Radiolabeling and Stability Assay

As shown in Figure 1A, the tracer [^18^F]**SF-AAN** was obtained by incubating nonradioactive probe [^19^F]**SF-AAN** with ^18^F in pyridazine-HCl buffer (pH 2.5) at 80 °C for 30 min. The tracer [^18^F]**SF-AAN** was generated with a high radiochemical conversion rate (>85%). After purification with a C18 light Sep-Pak cartridge, the radiochemical purity of [^18^F]**SF-AAN** was determined to be more than 99% (Figure 1B). Meanwhile, the molar activity of [^18^F]**SF-AAN** was calculated to be 12.77 ± 0.50 MBq/nmol (*n* = 5). This indicated that the tracer [^18^F]**SF-AAN** could be easily produced using a simple one-step radiofluorination method.

In order to meet the demand of biological applications, in vitro and in vivo stability of tracer [^18^F]**SF-AAN** was evaluated (Figure 1C). The radio-HPLC analysis distinctly displayed that no remarkable new peaks arose, even though [^18^F]**SF-AAN** was incubated in PBS at 37 °C for 4 h. This suggested that tracer [^18^F]**SF-AAN** was stable enough under physiological conditions. Furthermore, the radiochemical purity of [^18^F]**SF-AAN** in venous blood was still above 99% at 30 min post injection, indicating that it was also stable in vivo. These results supported that the radiotracer [^18^F]**SF-AAN** was sufficiently stable for preclinical study.

### 2.3. Reduction and Legumain-Controlled Self-Assembly

To confirm reducing environment and legumain-initiated intramolecular condensation, [^19^F]**SF-AAN** was used for the in vitro studies. As shown in Figure 2A, HPLC analysis indicated that after incubation with the reducing agent tris(2-carboxyethyl) phosphine (TCEP) at 37 °C for 30 min, the probe [^19^F]**SF-AAN** (retention time, 16.7 min) was converted into a new product [^19^F]**SF-AAN-R** (retention time, 14.7 min), which was verified by ESI-MS (Appendix A). Then, after [^19^F]**SF-AAN-R** was incubated with legumain for 1 h, new products [^19^F]**SF-R** and [^19^F]**SF-C** (retention time, 14.5 and 13.6 min) were generated and further verified by ESI-MS (Appendix A). All the results indicated that the probe [^19^F]**SF-AAN** could quickly form macrocycle [^19^F]**SF-C** in the presence of legumain and reducing agent.

Meanwhile, enzyme kinetics was also studied to analyze the cleavage efficiency of legumain to [^19^F]**SF-AAN**. *K_m_* and *K_cat_* of [^19^F]**SF-AAN** towards legumain were calculated using Michaelis−Menten equation. As shown in Figure 2B, the *K_m_* of [^19^F]**SF-AAN** was 103.10 μM, which was higher than the legumain-responsive fluorescent probe **1** (99.83 μM, log *P* = −0.07 ± 0.27) reported by our group previously [19]. This was presumably ascribed to the lipophilicity of probe [^19^F]**SF-AAN** (log *P* = 0.47 ± 0.03) affecting its interaction with legumain. Nevertheless, *K_cat_* (2.50 s^−1^) of [^19^F]**SF-AAN** was higher than that of the fluorescent probe **1** (1.8 s^−1^). The *K_cat_*/*K_m_* value of [^19^F]**SF-AAN** was measured to be 24,247.60 M^−1^ s^−1^, which was also higher than that of fluorescent probe **1** (18,030.65 M^−1^ s^−1^). This indicated that the probe [^19^F]**SF-AAN** designed in the present study could respond to legumain quickly.

### 2.4. Cellular Uptake Assay

According to the result of Western blot analysis (Figure 3A), human breast cancer cell line MDA-MB-468 with the highest expression level of legumain was chosen as the legumain-positive tumor cell model, and human prostate cancer cell PC-3 with the lowest expression level of legumain was chosen as the legumain-negative tumor cell model. Prior to investigating the targeting specificity of [^18^F]**SF-AAN**, the potential cytotoxicity against MDA-MB-468 and PC-3 cells was first evaluated using 3-(4,5-dimethyl-thiazol-2-yl)-2,5-diphenyltetrazolium bromide (MTT) assay, because an ideal imaging tracer should be nontoxic or low toxic. Even after incubating with [^19^F]**SF-AAN** (100 μM) for 24 h, the viability of MDA-MB-468 and PC-3 cells was still more than 95% (Appendix A). This verified that [^18^F]**SF-AAN** owned great biocompatibility and could be used for subsequent biological experiments.

Afterwards, the potential of [^18^F]**SF-AAN** for detecting legumain expression level in living MDA-MB-468 and PC-3 cells was investigated via cellular uptake assay. The group of PC-3 cells treated with [^18^F]**SF-AAN** displayed poor cellular uptake with the maximum value of only 1.89 ± 0.25% at 4 h (Figure 3B). However, the uptake of [^18^F]**SF-AAN** in MDA-MB-468 cells increased remarkably all the time, with the percentage of the total added radioactivity dose (AD%) value increasing from 2.22 ± 0.23% at 15 min to 3.76 ± 0.19% at 4 h, which was approximately two-fold more than that in PC-3 cells. Therefore, it was inferred that [^18^F]**SF-AAN** was capable of distinguishing legumain-overexpressed cancer cells specifically.

### 2.5. Intramolecular Condensation in Cancer Cells

At first, nonradioactive probe [^19^F]**SF-AAN** (50 μM) was incubated with MDA-MB-468 cell lysate at 37 °C for 10 h. Almost no intramolecular cyclized product was detected using HPLC (Figure 3C), which might attribute to the steric hindrance of disulfide bond [35] and a low glutathione (GSH) level in cell lysate. However, when [^19^F]**SF-AAN** was incubated with TCEP and MDA-MB-468 cell lysate at 37 °C for 10 h, disulfide bond in [^19^F]**SF-AAN** was reduced and the substrate of the probe [^19^F]**SF-AAN** was cleaved by legumain. Then, intramolecular cyclized product [^19^F]**SF-C** was formed. This verified that intramolecular condensation could carry out in MDA-MB-468 cells.

Similarly, after the radiotracer [^18^F]**SF-AAN** was incubated with MDA-MB-468 cells at 37 °C for 1 h, most of [^18^F]**SF-AAN** was converted into [^18^F]**SF-C**. Since the concentration of tracer [^18^F]**SF-AAN** was much less than that of nonradioactive probe [^19^F]**SF-AAN**, the disulfide bond in [^18^F]**SF-AAN** could be reduced quickly by the endogenous GSH. All the results demonstrated that [^18^F]**SF-AAN** could efficiently detect the expression level of legumain in living cancer cells. The formation of [^18^F]**SF-C** could effectively undergo self-assembly in situ for PET imaging of legumain activity in MDA-MB-468 cells.

### 2.6. Colocalization Assay

After incubation of cancer cells with [^19^F]**SF-AAN** for 6 h, the fluorescence intensity of the probe [^19^F]**SF-AAN** in MDA-MB-468 cells was higher than that in PC-3 cells (Figure 4A), which was consistent with the result of cellular uptake. To further investigate the distribution of [^19^F]**SF-AAN** in cancer cells, MDA-MB-468 and PC-3 cells were incubated with [^19^F]**SF-AAN** and lysosome tracker. Confocal microscopy imaging was applied to detect the subcellular location of [^19^F]**SF-AAN**. In MDA-MB-468 cells, the blue fluorescence of probe [^19^F]**SF-AAN** colocalized well with the red fluorescence produced by the lysosome tracker (Figure 4B). As a result, when the probe [^19^F]**SF-AAN** was internalized into MDA-MB-468 cells, it could be activated in response to the overexpressed legumain in the lysosome. On the contrary, the fluorescence of [^19^F]**SF-AAN** was quite low in PC-3 cells due to the low expression level of legumain in PC-3 cells.

### 2.7. PET Imaging of Tumor-Bearing Mice

Prior to PET imaging, pharmacokinetics was studied for the radiotracer [^18^F]**SF-AAN**. The elimination half-life (t_1/2z_) of [^18^F]**SF-AAN** was determined to be 50.05 min, indicating that tracer [^18^F]**SF-AAN** could be rapidly cleared from the blood (Appendix A). On one hand, rapid clearance of [^18^F]**SF-AAN** in blood could reduce the radioactive damage to normal organs. On the other hand, rapid clearance of tracer was conducive to promoting the SNR.

Encouraged by the above results, the ability of [^18^F]**SF-AAN** for detecting legumain activity in vivo was evaluated in MDA-MB-468 and PC-3 tumor-bearing mice. As shown in Figure 5A,B, the tumor uptake of [^18^F]**SF-AAN** in MDA-MB-468 tumor-bearing mice was significantly higher than that in the PC-3 tumor-bearing mice at all the time points. The radiotracer [^18^F]**SF-AAN** could quickly arrive at the tumor site in MDA-MB-468 tumor-bearing mice at 10 min post injection and the tumor uptake reached the maximum of 4.22 ± 0.40% ID/mL (Figure 5A,C), which was about 3 times higher than that of muscle (Figure 5E). As time went on, the tumor uptake decreased rapidly. At 60 min post injection, the tumor uptake decreased to 1.63 ± 0.07% ID/mL, but the SNR was still as high as 2.26 ± 0.18. Compared with the tracer **^18^F-2** reported previously, [^18^F]**SF-AAN** could detect legumain expression level more sensitively and specifically, which did not need co-injection with additional nonradioactive probe to obtain ideal PET imaging.

Oppositely, the maximum tumor uptake was only 1.56 ± 0.37% ID/mL in PC-3 tumor-bearing mice at 10 min post injection, which was similar to the muscle uptake (Figure 5B,D). In addition, the PC-3 tumor uptake decreased from 1.56 ± 0.37% ID/mL at 10 min to 1.03 ± 0.22% ID/mL at 60 min. The maximum SNR was determined to be 1.66 ± 0.25 at 20 min and decreased to 1.12 ± 0.23 at 60 min. In consequence**,** this demonstrated that [^18^F]**SF-AAN** could be utilized as a new PET tracer to immediately visualize endogenous legumain. In this study, [^18^F]**SF-AAN** could identify tumors with disparate legumain expression level sensitively and specifically.

The specificity of [^18^F]**SF-AAN** to legumain was further assessed by autoradiography ex vivo. As shown in Figure 6A, the tracer [^18^F]**SF-AAN** possessed great tissue penetration ability and targeting specificity. The radioactivity in MDA-MB-468 tumor was 7.7 times of that in muscle (Figure 6B). On the contrary, the radioactivity in PC-3 tumor was only 3 times of that in muscle, which was dramatically lower than that in MDA-MB-468 tumor. All the results were fundamentally in accordance with the results of PET imaging.

### 2.8. Biodistribution and Histopathological Analysis

The biodistribution of [^18^F]**SF-AAN** in MDA-MB-468 tumor-bearing mice was further studied and shown in Appendix A. The accumulation of tracer [^18^F]**SF-AAN** in liver, kidney and small intestine was relatively high at 60 min post injection, which was fundamentally consistent with the results of PET imaging. This is mainly due to the fact that the liver and kidney were considered as the main metabolic organs and legumain was also expressed in these organs [12]. The tumor uptake was determined to be 1.58 ± 0.08% ID/g and 1.02 ± 0.09% ID/g in MDA-MB-468 tumor-bearing mice at 30 min and 60 min post injection, respectively. On the contrary, the muscle uptake was only 0.61 ± 0.01% ID/g and 0.43 ± 0.16% ID/g at 30 min and 60 min post injection, respectively. The difference of uptake was attributed to the high legumain expression in MDA-MB-468 tumor. In addition, major organs and tumor displayed no obvious pathological change (Appendix A), indicating that the tracer [^18^F]**SF-AAN** owned great biocompatibility.

## 3. Discussion

As the incidence of cancer is increasing year by year and legumain is highly expressed in cancers, our group designed and synthesized a smart PET tracer **^18^F-2** to selectively visualize the legumain-positive tumors for cancer diagnosis, but the tumor uptake was not high and nonradioactive probe needed to be co-injected to obtain ideal tumor uptake [24]. Therefore, we further optimized the structure of tracer to get a novel legumain-responsive tracer [^18^F]**SF-AAN** for sensitively and accurately detecting legumain activity. The possible mechanism of action was put forward for the legumain-triggered PET imaging tracer [^18^F]**SF-AAN**, as described in Figure 1. Primarily, the disulfide bond of tracer [^18^F]**SF-AAN** could be cleaved by endogenous GSH in the cancer cells to form [^18^F]**SF-AAN-R**. Subsequently, [^18^F]**SF-AAN-R** would enter into lysosome and the **AAN** sequence could be quickly cleaved by activated legumain. At the same time, the sulfhydryl group and amino group of cysteine were exposed, and hence a bioorthogonal condensation [36,37,38,39] took place with the 2-cyano group of CBT to form the intramolecular cyclized compound [^18^F]**SF-C**. Since [^18^F]**SF-C** was more rigid and hydrophobic, the self-assembly of [^18^F]**SF-C** in situ was promoted via intermolecular π-π stacking interactions. The self-assembled aggregates could generate a strong radioactive signal, which is beneficial for PET imaging legumain activity in cancer cells or tumor-bearing mice sensitively and specifically.

The tracer [^18^F]**SF-AAN** was obtained with relatively high radiochemical conversion rate and purity. More importantly, the in vivo stability of tracer was significantly improved to meet the standard of biological applications. In vitro experiments demonstrated that the probe could be activated by GSH and legumain to form an intramolecular cyclized compound. This definitely verified the mechanism of reduction and legumain-controlled signal amplification for imaging the activity of legumain. The longer retention and higher cellular uptake of [^18^F]**SF-AAN** in MDA-MB-468 cells may be attributed to the intramolecular condensation of [^18^F]**SF-AAN** triggered by legumain in legumain-positive cells. The hydrophobic intramolecular cyclized product could further self-assemble into aggregates for prolonging the retention time of [^18^F]**SF-AAN** in MDA-MB-468 cells. Moreover, intramolecular cyclized product [^18^F]**SF-C** could be detected in MDA-MB-468 cancer cells, which further demonstrated the specific response of [^18^F]**SF-AAN** toward legumain activity. Confocal studies clearly displayed the potential of [^18^F]**SF-AAN** for distinguishing legumain overexpressed cancers.

Our study displayed that MDA-MB-468 tumor-bearing mice, which owned high legumain expression, could be clearly visualized with high tumor uptake after the injection of [^18^F]**SF-AAN**, while PC-3 tumor-bearing mice showed almost background uptake in vivo at all the time point. A remarkable increase in radioactive signal could be detected in legumain-positive MDA-MB-468 tumor within 10 min and SNR was also remarkably higher than that of legumain-negative PC-3 tumors. This illustrated that the tumor uptake of [^18^F]**SF-AAN** was positively associated with the expression level of legumain. Compared with **^18^F-2**, [^18^F]**SF-AAN** could detect legumain activity more sensitively and specifically without co-injection with nonradioactive probe [^19^F]**SF-AAN**. Meanwhile, the results of autoradiography further verified that radiotracer [^18^F]**SF-AAN** could assess the expression level of legumain in cancer at in vivo level specifically and sensitively. Nevertheless, high radioactivity accumulation in liver, kidney and small intestine was observed. This was due to the fact that the tracer was lipophilic and these organs were main metabolic organs, where legumain was also endogenously expressed. Moreover, no obvious pathological change was found according to H&E staining, which further proved that tracer [^18^F]**SF-AAN** had good biocompatibility. Hence, [^18^F]**SF-AAN** could provide assistance for clinical treatment and even predict the malignant degree of cancer. In the near future, further optimization of [^18^F]**SF-AAN** would be made to reduce the tracer uptake in non-target organs particularly in the liver and enhance the retention of the tracer at the tumor site.

## 4. Materials and Methods

### 4.1. General Information

All chemical reagents and solvents were obtained from commercial sources (Energy Chemical, Shanghai, China) and used without additional purification except as otherwise noted. CBT was purchased from Subo Chemical Technology Company Limited (Shanghai, China). The ethylenediaminetetraacetic acid and dithiothreitol applied for enzyme kinetic studies were purchased from Sangon Biotech (Shanghai, China). Human breast cancer cell lines MDA-MB-231 and MDA-MB-468, human prostate cancer cell line PC-3 and human colon cancer cell line RKO were obtained from the cell bank of Chinese Academy of Sciences (Shanghai, China). HPLC was composed of a pump (Waters 1525 HPLC, Waters, Singapore) and connected with a reverse phase column (RP-C18, 4.6 × 250 mm, 10 μm, Elite Analytical Instrument Company, Dalian, China), a UV detector (2998 multi-wavelength absorbance, Waters, Milford, MA, USA) as well as a radioactivity detector (ElySia Raytest, Straubenhardt, Germany). A MeCN/H_2_O gradient mobile phase including 0.1% trifluoroacetic acid (TFA) was used for analysis or purification (at a flow rate of 1 or 3 mL/min). HPLC conditions for analysis and purification of the compounds were listed in Appendix A. ESI-MS was obtained through a quadrupole tandem mass spectrometer (SQ-detect 2) (Waters, Milford, MA, USA). ^1^H-NMR and ^13^C-NMR spectra were recorded on a Bruker DRX-400 spectrometer (Ettlingen, Germany). Fluorine-18 was gained with the medical cyclotron (Sumitomo HM-7, Tokyo, Japan). The radioactivity of tracer [^18^F]**SF-AAN** was counted with a Gamma counter (2470, Perkin-Elmer Corporation, Waltham, MA, USA). Cell images were obtained on the Olympus Fluoview 500 IX71 confocal microscope (Tokyo, Japan).

### 4.2. Synthesis of Nonradioactive Probe [^19^F]**SF-****A****AN**

To a solution of compound **SF-11** (14.53 mg, 0.0192 mmol) in tetrahydrofuran (THF, 3 mL) and N,N-Dimethylformamide (DMF, 0.5 mL), 2-(1Hbenzotriazole-1-ly)-1,1,3,3-tetramethyluronium hexaflourophosphate (HBTU, 8.4 mg, 0.022 mmol), **A****c****-AAN-OH** (12 mg, 0.0211 mmol), and diisopropylethylamine (DIPEA, 9 µL, 0.048 mmol) and were added. Then, the reaction mixture was stirred at 25 °C under N_2_ for 3 h and the product was evaporated under reduced pressure to yield the compound **SF-12**. In order to remove the protecting group of Trt, triisopropylsilane (Tips, 40 µL) and TFA (2 mL) were added to a solution of compound **SF-12** (24.91 mg, 0.0192 mmol) in dichloromethane (2 mL), which was stirred at 25 °C for 0.5 h. Then the solvent was evaporated under the reduced pressure and coarse product was precipitated from the cold diethyl ether followed by centrifugation to obtain the compound **SF-13**. The compound **SF-13** (20.25 mg, 0.0192 mmol), AMBF_3_ (150 µL, 100 mg/mL), ligand (194 µL, 4 mg/mL) and tetrakis(acetonitrile)copper(I) hexafluorophosph (8 mg, 0.0215 mmol) was dissolved in a solution of DMF and H_2_O (2/1, *v*/*v*), which was stirred under N_2_ at 45 °C for 45 min. The rough product was purified by preparative HPLC to obtain pure nonradioactive probe [^19^F]**SF-AAN** (11 mg, 45.8%). ^1^H NMR (400 MHz, DMSO-*d*_6_) δ 10.73 (s, 1H), 9.07 (t, *J* = 6.0 Hz, 1H), 8.80 (d, *J* = 7.5 Hz, 1H), 8.74 (d, *J* = 2.0 Hz, 1H), 8.28 (d, *J* = 7.8 Hz, 1H), 8.22–8.18 (m, 2H), 8.13–8.02 (m, 4H), 7.85 (dd, *J* = 8.1, 4.3 Hz, 4H), 7.52 (s, 1H), 7.41 (d, *J* = 7.9 Hz, 2H), 7.34 (d, *J* = 8.0 Hz, 2H), 4.91 (t, *J* = 7.5 Hz, 1H), 4.84 (t, *J* = 6.9 Hz, 2H), 4.54–4.51 (m, 2H), 4.36 (d, *J* = 6.0 Hz, 4H), 3.82 (d, *J* = 6.1 Hz, 1H), 3.78 (d, *J* = 6.0 Hz, 1H), 3.69 (d, *J* = 2.8 Hz, 2H), 3.31–3.23 (m, 2H), 2.89 (s, 2H), 2.73 (s, 2H), 2.69 (t, *J* = 7.2 Hz, 2H), 2.56 (t, *J* = 5.5 Hz, 2H), 2.36 (q, *J* = 4.7 Hz, 2H), 1.83 (s, 3H), 1.23 (d, *J* = 7.3 Hz, 3H), 1.20 (d, *J* = 3.0 Hz, 3H), 1.18 (d, *J* = 2.6 Hz, 3H). ^13^C NMR (101 MHz, DMSO-*d*_6_) δ 172.94, 172.73, 171.12, 169.65, 169.09, 166.92, 166.57, 162.77, 148.18, 143.88, 143.21, 139.82, 135.56, 132.72, 128.07, 127.72, 127.34, 127.30, 125.22, 124.29, 121.44, 114.03, 112.07, 63.87, 54.95, 53.44, 52.97, 50.19, 49.06, 48.79, 48.60, 43.98, 42.96, 42.17, 37.23, 36.25, 34.80, 32.15, 31.23, 22.92, 18.46, 18.34, 14.75.

### 4.3. Radiosynthesis of Tracer [^18^F]**SF-AAN**

The tracer [^18^F]**SF-AAN** was obtained by using a simple one-step labeling method [40]. The isotope ^18^F was acquired with a QMA column and then eluted into a tube with pyridazine-hydrochloride buffer (350 μL, pH 2.5). Nonradioactive probe [^19^F]**SF-AAN** in DMF (25 mM, 20 μL) was added to the buffer and incubated at 80 °C for 30 min. Subsequently, the mixture was transferred to water (20 mL) and the tracer [^18^F]**SF-AAN** was captured with a C18 light Sep-Pak cartridge. The superfluous ^18^F was removed with water (30 mL) and the final ^18^F-labeled tracer [^18^F]**SF-AAN** was eluted with ethanol (500 μL). Before and after purification, a little sample was taken to determine the radiochemical conversion rate and purity of [^18^F]**SF-AAN** with radio-HPLC.

### 4.4. Stability Assay

The stability of tracer [^18^F]**SF-AAN** in vitro was assessed by incubating [^18^F]**SF-AAN** (20 μL, 11.1 MBq, 0.30 mCi) with PBS (90 μL, pH 7.4) at 37 °C for 1, 2 and 4 h, respectively. Upon reaching the time point, 20 μL of the solution was taken out for monitoring the radiochemical purity change of tracer [^18^F]**SF-AAN** by radio-HPLC.

In vivo stability assay was performed by injecting the tracer [^18^F]**SF-AAN** (22.2 MBq, 0.60 mCi) diluted with saline into mouse via the tail vein. After 30 min, blood was collected from the tail vein of nude mouse, and then acetonitrile (50 μL) was added to precipitate protein, which was centrifuged at 12,000× *g* for 5 min. Subsequently, the supernatant was collected for determining the radiochemical purity by radio-HPLC to evaluate in vivo stability.

### 4.5. Measurement of Water Partition Coefficient

The water partition coefficient (log *P*) of [^18^F]**SF-AAN** was determined by measuring the radioactivity of tracer [^18^F]**SF-AAN** in water and n-octanol with a Gamma counter. The log *P* was calculated with the following formula, log *P* = log (*C_o_*/*C_w_*), where *C_o_* denoted the radioactivity of [^18^F]**SF-AAN** in the organic phase and *C_w_* denoted the radioactivity of [^18^F]**SF-AAN** in the aqueous phase.

### 4.6. Reduction and Legumain-Controlled Self-Assembly

A solution of nonradioactive probe [^19^F]**SF-AAN** (50 μM) in DMF was incubated with TCEP (1 mM) at 37 °C for 30 min. Afterwards, legumain working buffer (1 μg/mL of legumain, 20 mM of citric acid, 60 mM of disodium phosphate, 1 mM of EDTA, 1 mM of DTT, pH of 5.5) was added to the above solution, which was incubated at 37 °C for another 1 h. Then 20 μL of the solution was collected for HPLC analysis.

In order to further assess the legumain-triggered cleavage efficiency of the probe, enzyme kinetics studies were carried out for probe [^19^F]**SF-AAN**. Different concentrations of [^19^F]**SF-AAN** (50, 100, 150, 200, 300 and 500 μM) were incubated with TCEP (1, 2, 3, 4, 6 and 10 mM) at 37 °C for 30 min, respectively, and then incubated with legumain working buffer at 37 °C for 60 min. The hydrolysis efficiency of legumain toward different concentrations of [^19^F]**SF-AAN** was detected with HPLC at 320 nm. The kinetic parameters were calculated with the following the Michaelis–Menten equation,
$$v=\frac{V_{max}\;\left[S\right]}{K_{m}+\left[S\right]}$$
where *V_max_* denoted the maximum shearing rate; *K_m_* denoted the substrate concentration when the hydrolysis efficiency arrived at the half of *V_max_*; and [*S*] was the concentration of probe.

### 4.7. Cell Culture and Western Blot Analysis

MDA-MB-231, PC-3 and RKO cells were seeded in RPMI-1640 medium including 10% fetal bovine serum (FBS) and cultured at 37 °C with 5% CO_2_ in an artificial humid atmosphere. However, MDA-MB-468 cells were cultured in the Leibovitz’s L-15 medium containing 10% FBS at 37 °C without 5% CO_2_ in a humidified atmosphere. The medium was replaced every other day to maintain cells continuous and desired growth. To ascertain the expression level of legumain in different cells (MDA-MB-231, MDA-MB-468, PC-3 and RKO), Western blot analysis was carried out. Firstly, the cells were cultured for 24 h and then lysed with RIPA (Beyotime Biotechnology, Shanghai, China) at 4 °C. Secondly, the same quantity of protein of cell lysate (50 μg) was separated on a 10% of SDS-polyacrylamide gel electrophoresis (SDS-PAGE). Thirdly, protein bands were transferred to polyvinylidene difluoride (PVDF) transfer membrane for about 1 h. Then, the membrane was blocked with 5% skimmed milk for 1 h followed by washing with TBST buffer three times. Then, the membrane was incubated with anti-legumain antibody (1:750, Abcam plc.) or anti-β-actin (1:1000) at 4 °C overnight. The bands were washed three times and incubated with rabbit or goat anti-mouse IgG-HRP (1:10,000, Santa Cruz) at 25 °C for 1 h. Finally, the protein bands were visualized using an ECL Western blot kit, and densitometry analysis of the film was performed with ImageJ software.

### 4.8. In Vitro Biocompatibility Test

The biocompatibility of nonradioactive probe [^19^F]**SF-AAN** was studied with MTT assay. MDA-MB-468 and PC-3 cells were cultured in a 96-well plate at a density of 1 × 10^4^ cells/well. After 24 h, medium was removed and the fresh medium containing [^19^F]**SF-AAN** (0, 10, 20,40, 80 and 100 μM) was added to each well, which was cultured for another 24 h. Afterwards, each well was added with MTT (5 mg/mL, 20 μL) and the cells were incubated for additional 4 h. With the purpose of dissolving the purple crystals, after medium was removed, DMSO (150 μL) was added to each well and 96-well plate was shaken for 10 min. The absorbance of each well at 490 nm was measured by an ELISA reader (BioTek uQuant, Winooski, VT, USA) and the cell viability was determined from the OD value of the experimental group to that of the control group The biocompatibility of tracer [^18^F]**SF-AAN** was evaluated accordingly.

### 4.9. Cellular Uptake Assay

Western blot analysis suggested that MDA-MB-468 cells expressed high level of legumain, while PC-3 cells expressed low level of legumain. Hence, MDA-MB-468 and PC-3 cells were chosen as legumain-positive and negative models to test the targeting specificity of [^18^F]**SF-AAN**. The tracer [^18^F]**SF-AAN** dissolved in serum-free medium was added to tubes containing 1 × 10^7^ cells (37 kBq, 300 μL), which was incubated at 37 °C for 15, 30, 60, 120 and 240 min, respectively. At each specified time point, the cells were washed and centrifuged with cold PBS (500 μL) twice to remove unabsorbed radioactive tracer. Finally, the radioactivity of [^18^F]**SF-AAN** which was uptaken by cells was measured via a Gamma counter.

### 4.10. Intramolecular Condensation in Cancer Cells

For evaluating the self-assembly ability of probe in MDA-MB-468 cells, nonradioactive probe [^19^F]**SF-AAN** (0.2 μL, 25 mM) was firstly added to MDA-MB-468 cell lysate (100 μL) co-incubated with or without 1 mM TCEP at 37 °C for 10 h. After centrifugation of mixture, supernatant (20 μL) was taken for HPLC analysis to detect the formation of [^19^F]**SF-C**.

In addition, the tracer [^18^F]**SF-AAN** (22.2 MBq, 0.60 mCi) was also incubated with MDA-MB-468 cells (3 × 10^7^) at 37 °C for 1 h. Afterwards, DMSO (100 μL) was added to lyse the cells, which was centrifuged at 12,000× *g* for 5 min. Then, the supernatant was collected for detecting the formation of [^18^F]**SF-C** by radio-HPLC.

### 4.11. Colocalization Assay

MDA-MB-468 and PC-3 cells were cultured in a 96-well black plate at the density of 2 × 10^4^ cells/well in medium (100 μL) for 24 h. Subsequently, the medium was removed and the solution of [^19^F]**SF-AAN** (0, 10, 20, 40, 80 and 100 μM) was added to each well and incubated for 6 h. Then, the cells were washed with PBS and added with PBS (200 μL/well). The fluorescence intensity of each well was measured under the excitation of 340 nm and emission of 415 nm using a ThermoFisher Scientific Varioskan Flash microplate reader (USA).

For colocalization studies, MDA-MB-468 and PC-3 cells were incubated with nonradioactive probe [^19^F]**SF-AAN** (1.5 mL, 50 μM) for 6 h, followed by removing medium and incubating with the lysosome tracker (Red DND-99, 1.5 mL, 1 μM) at 37 °C for 10 min. Subsequently, the medium was removed and the cells were washed with PBS three times. Finally, the fluorescence image was recorded by a confocal microscope, which was magnified 400 times and 1 × 10^3^ times, respectively.

### 4.12. Pharmacokinetics Study

The tracer [^18^F]**SF-AAN** (5.55 MBq, 0.15 mCi) diluted with saline (150 μL) was injected into normal mice intravenously (*n* = 3). Venous blood was collected at 1, 5, 7, 10, 15, 20, 30, 45, 60, 90 and 120 min, respectively. Then, each sample was weighed and radioactivity in each sample was measured with a Gamma counter. Drug and Statistics 2.1 (DAS 2.1) software was applied to study the pharmacokinetics of tracer [^18^F]**SF-AAN** in vivo.

### 4.13. In Vivo PET Imaging

BALB/c nude mice (female, 4–5 weeks old, 16–20 g) were purchased from Changzhou Cavens Laboratory Animal Co. Ltd. The mice were subcutaneously injected with MDA-MB-468 and PC-3 cells (1 × 10^8^ and 5 × 10^7^ in 50 μL PBS) into the right flank. All the tumor-bearing mice were cultured in a sterile room for about 3–4 weeks. When the diameter of the tumor reached 0.7–0.8 cm, the tumor-bearing mice could be used for PET imaging.

Before PET imaging, the mice were kept fasting for 6 h. The tracer [^18^F]**SF-AAN** (5.55 MBq, 0.15 mCi) was injected to tumor-bearing mice via the tail vein for the PET imaging. Dynamic PET imaging was performed for the first 60 min utilizing an Inveon scanner (Siemens Inven, Berlin, Germany). During the entire PET scanning process, mice were under anesthesia with isoflurane gas (1.5~2.0%). The tracer uptake in tumor and other tissues was quantified using the region of interest (ROI) technique, which was expressed as percent of injected dose per cubic centimeter (% ID/mL). Autoradiography analysis was also carried out to determine and compare the uptake of [^18^F]**SF-AAN** in the tumor and muscle of MDA-MB-468 and PC-3 tumor-bearing mice.

### 4.14. Ex Vivo Biodistribution and Histopathological Analysis

The tracer [^18^F]**SF-AAN** (5.55 MBq, 0.15 mCi) was injected into MDA-MB-468 tumor-bearing mice intravenously. At 30 min and 60 min post injection, mice were dissected, and the main organs were taken out and weighed. Meanwhile, radioactivity in each organ was detected with a Gamma counter. The value was expressed as the percent of injected dose per gram of tissue (% ID/g).

In addition, hematoxylin and eosin (H&E) staining was also performed to monitor the uptake of tracer [^18^F]**SF-AAN** in the tumor and other organs, such as heart, liver, spleen, lung, kidney, intestine, and stomach. Each organ was put in 15% and 30% sucrose solutions to dehydrate at 4 °C overnight. Afterwards, organs were frozen and sliced into the section thickness of 6 μm. Afterwards, all slices were placed in a 37 °C drying oven overnight. Then H&E staining kit (Beyotime) was applied to stain the tissue sections. Finally, images were obtained via fluorescence microscope (OLYMPUS IX51, Tokyo, Japan).

### 4.15. Statistical Analysis

All data were presented as mean ± standard deviation (SD). Origin 2018 was applied for the statistical analysis. *T*-test was used for comparison between two groups. In all statistical analysis, N.S. denoted no statistical significance, * *p* < 0.05, ** *p* < 0.01 and *** *p* < 0.001.

## 5. Conclusions

In summary, a novel fluorine-18 labeled tracer [^18^F]**SF-AAN** was designed for detecting legumain activity in vivo sensitively and specifically. It was obtained using a convenient one-step ^18^F-^19^F isotope exchange method with high radiochemical conversion rate, purity and stability. In the presence of GSH and legumain, reduction and legumain-controlled intramolecular condensation of nonradioactive probe [^19^F]**SF-AAN** to form [^19^F]**SF-C** was validated. Cellular uptake demonstrated that the uptake of the tracer [^18^F]**SF-AAN** in MDA-MB-468 cells was particularly higher owing to legumain overexpression. Intramolecular cyclized product [^18^F]**SF-C** could be formed for the detection of legumain activity in MDA-MB-468 cells. In addition, in vivo PET imaging and biodistribution studies showed that [^18^F]**SF-AAN** could detect legumain expression level in the tumor accurately and rapidly. The legumain-triggered bioorthogonal macrocyclization and aggregation of tracer allowed for the effective diagnosis of cancer in vivo. We anticipate that [^18^F]**SF-AAN** could be applied for the detection of legumain level to diagnose cancer after further optimization.

## Data Availability

Data are contained within the article and supplementary material.

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
