# Peer review of "Development of a Promising 18F-Radiotracer for PET Imaging Legumain Activity In Vivo"

_pharmaceuticals, 2022, doi:10.3390/ph15050543_

Round 1

Reviewer 1 Report

I have revised the manuscript entitled "Development of a promising 18F-radiotracer for PET imaging legumain activity in vivo" by Lu and co-authors (Manuscript #pharmaceuticals-1608008), which has been submitted for publication in Pharmaceuticals. Here, the editor and authors can find my report:

In the present work, the authors designed and synthesized a 18F-labeled tracer ([18F]SF-AAN) targeting the legumain activity overexpressed in tumors. The work brings the idea of increased enzyme activity as a potential target for tumor detection using potential radiolabeled probes, such as the one proposed in this research. I think this is an interesting approach in the development of new radiopharmaceuticals for tumor detection and evaluation. So any contribution that advances this area is of high interest in the radiopharmaceutical and nuclear medicine fields.

Considering all these together, the manuscript is suitable for the scope of Pharmaceuticals. Some minor revisions must be addressed by the authors before final publication:

  1. During my revision, I had no access to the supplemental files. Attention to the submission of these data, which facilitate the general understanding of the proposed chemical modifications in the aforementioned molecule (SF-11 / 18F-2).

  1. As the style of Pharmaceuticals is to provide the “Results” section before the “Materials and methods” section, the authors should carefully revise some abbreviations, which were used in the “Results” section, but only described in the “Materials and methods” section. Abbreviations should be described at the first time they are used in the main text, which, in this case, is the “Results” section.

  1. Was the tracer [18F]SF-AAN based on the SF-11 compound (as stated in line 72; last paragraph of the “Introduction” section) or on the 18F-2 compound (as stated in line 82; section 2.1)? Supplemental files were not available.

  1. What is the meaning of AMBF3 motif (line 88; section 2.1)?

  1. Could the authors comment on how the lipophilicity of [19F]SF-AAN affected its interaction with legumain, as stated in the second paragraph of the section 2.3?

  1. Section 2.4: What does “AD%” mean?

  1. Could the authors comment on the time for imaging acquisition within 60 min for 18F-labeled molecules and “the elimination half-life (t1/2z) of [18F]SF-AAN was determined to be 50.05 min, indicating that tracer [18F]SF-AAN could be rapidly cleared from the blood at 2 h post injection”?

  1. “Discussion” section (paragraph 1): A reference should be included after “a smart PET tracer 18F-2 to selectively visualize the legumain-positive tumors for cancer diagnosis, but the tumor uptake was not high and nonradioactive probe needed to be co-injected to obtain ideal tumor uptake.”

  1. In the discussion section (paragraph 2), the authors state that “The hydrophobic intramolecular cyclized product could further self-assemble into aggregates for prolonging the retention time of [18F]SF-AAN in MDA-MB-468 cells”. However, the radiotracer was rapidly cleared out from the tumor, as stated by the authors in the section 2.7 (“As time went on, the tumor uptake decreased rapidly”). I think authors could better discuss this issue. I do not agree that the “self-assemble into aggregates for prolonging the retention time”.

  1. Discussion section (paragraph 3): “A remarkable increase of radioactive signal was detected for MDA-MB-468 tumor within 10 min on account of its fast half-life of distribution.” and “An obvious accumulation of tracer could be detected in MDA-MB-468 tumors with high legumain expression at first 10 min, and SNR was remarkably higher than legumain negative PC-3 tumors.” sound redundant.

  1. Section 5.4: How much blood was collected from the mouse? What was the proportion between blood and acetonitrile?

  1. Verify whether the concentration of [19F]SF-AAN (50 µM) in the section 2.5 corresponds to the respective method section 5.10.

  1. Section 5.14: After H&E staining, images of tumor and the organs were obtained in a fluorescence microscope???

  1. Section 5.15: “Standard error” or “standard deviation”? For the comparison between two groups were used both t-test and one-way ANOVA? The latter is suitable for the comparison between three or more groups.

I would like to congratulate the authors for their excellent research.

Reviewer 2 Report

Please see attached file for comments.

Round 2

Reviewer 2 Report

I thank the authors for answering my question. And I don't have further comments for the paper.